



# Quantifying the impact of the structural uncertainty on the gross rock volume in the Lubina and Montanazo oil fields (Western Mediterranean)

Carla Patricia Bárbara[1], Patricia Cabello[2*], Alexandre Bouche[3], Ingrid Aarnes[4], Carlos Gordillo[5], Oriol Ferrer[2], Maria Roma[2], Pau Arbués[2]

[1] Independent researcher, Nottingham, United Kingdom.
[2] Dept. Dinàmica de la Terra i de l'Oceà. Institut de Recerca Geomodels. Facultat de Ciències de la Terra, Universitat de Barcelona, c/Martí i Franquès s/n, 08028, Barcelona, Spain
[3] Emerson Automation Solutions, Lysaker Torg 45, 1366, Lysaker, Norway
[4] Norwegian Computing Center, Gaustadalleen 32a, NO-0373 Oslo, Norway
[5] Repsol, Repsol Campus Madrid, Blue Building, c/Méndez Álvaro 44, 1st floor, 28045, Madrid, Spain

*Correspondence to*: Patricia Cabello (pcabello@ub.edu)

**Abstract.** Structural uncertainty is a key parameter affecting the accuracy of the information contained in static and dynamic reservoir models. However, quantifying and assessing its real impact on reservoir property distribution, in-place volume estimates and dynamic simulation has always been a challenge. Due to the limitation of the existing workflows and time constraints, the exploration of all potential geological configurations matching the interpreted data has been limited to a small number of scenarios, making the future field-development decisions uncertain.

We present a case study in the Lubina and Montanazo mature oil fields (Western Mediterranean) in which the structural uncertainty in the seismic interpretation of faults and horizons has been captured using modern reservoir modeling workflows. We model the fault and horizon uncertainty by means of two workflows, the manually interpreted and the constant uncertainty cases. In the manually interpreted case, the zones of ambiguity in the position of horizons and faults are defined as locally varying envelopes around the best interpretation, whose dimensions vary according to the diffractions and amplitudes of the seismic data throughout the surface interpretation. In the constant case, the envelope dimensions are kept constant for each horizon and each fault. Both faults and horizons are simulated within their respective uncertainty envelopes as provided to the user. In all simulations, conditioning to available well data is ensured. Stochastic simulation was used to obtain 200 realizations for each uncertainty modeling workflow. The realizations were compared in terms of gross rock volumes above the oil-water contact considering three scenarios in the depths of the contact.

The results show that capturing the structural uncertainty in the picking of horizons and faults in seismic data has a relevant impact on the volume estimation. The models predict percentage differences in the mean gross rock volume with respect to best estimate interpretation up to 16% higher and 22% lower. The manually interpreted uncertainty workflow reports narrower gross rock volume predictions and more consistent results from the simulated structural models than the constant case. This work has also revealed that, for the Lubina and Montanazo fields, the fault uncertainty associated with the major faults that bound laterally the reservoir strongly affect the GRV predicted. The multiple realizations obtained are geologically consistent





with the available data and their differences in geometry and dimensions of the reservoir allows us to improve the understanding of the reservoir structure.

The uncertainty modeling workflows applied are easy to design and allow to update the models when required. This work demonstrates that knowledge of the data and the sources of uncertainty is important to set up the workflows correctly. Further

studies can combine other sources of uncertainty in the modeling process to improve the risk assessment.

# 1 Introduction

Geological modeling is a powerful tool that allows us to obtain realistic representations of the subsurface, which in turn gives a better understanding of the most geologically complex scenarios (e.g. Jolie et al., 2015; Hoffman et al., 2008; Latief et al., 2012). Geological modeling has also become a key task used for the correct management of georesources, such as

hydrocarbons, mineral and geothermal resources, and water management, where it is used to make decisions and to perform risk assessments. Geological modeling is also applied in public works and civil engineering.

Models describing the structure of the subsurface are built from the best estimate interpretation of data. Due to software limitations and time constraints, only a limited number of possible modeling scenarios are produced by the interpreters. This practice, which ignores the other equally probable solutions that match all the available data and interpretations, has often led

to unpleasant surprises, for example, when new wells have been drilled. Another challenge is that many of these models are not easy to update with new data when it becomes available (Seiler et al., 2009; Skjervheim et al., 2012; Pettan and Strømsvik, 2013), and companies may run the risk of taking decisions based on models that are no longer valid.

A geological model of the subsurface should capture the key uncertainties relevant for the modeling task in question. Most modeling workflows are to a large degree acknowledging the uncertainty associated with the input data when it comes to

populating the 3D reservoir grids with static and dynamic properties (e.g. Cabello et al., 2011, 2018). However, many of these workflows operates with a locked geometry of the 3D grid, and thus are not able to capture the often highly significant structural uncertainties. The aim of incorporating all significant uncertainties is to improve the predictive power of the models.

In particular, the uncertainty related to the geological structure of the subsurface is considered to produce large impact on outcomes obtained from the geological model. The structural uncertainties are relevant for the most common reservoir

modeling aims, such as static volume estimates, dynamic flow simulations and well planning and drilling operations. The structural uncertainty is affected by the quality, resolution and spatial distribution of the input data (Wellmann and Regenauer-Lieb, 2012; Bond, 2015). These uncertainties should be propagated through each step in the subsequent modeling workflow (MacDonald et al., 2009; Suslick et al., 2009; Neumann et al., 2012). Such workflows needs to be fully automated in order to provide rapid and efficient results.

Various sources of structural uncertainty exist, including conceptual, geophysical and well data uncertainties. Conceptual uncertainty responds to the inherent uncertainty in the interpretation of data that mainly comes from human bias and depends



on the expertise of the interpreter (Bond et al., 2007; Rojas et al., 2010; Bond, 2015; Alcalde et al., 2017a, b; Howley and Meyer, 2015). It explains differences in interpretations by different geoscientists that result from the same input data.

Geophysical uncertainties associated with seismic data affecting a structural model are those resulting from migration, time picking (interpretation) and time-to-depth conversion (Thore et al., 2002). The uncertainty associated with the seismic

interpretation of horizons and faults is inherent to the low resolution and quality of seismic data. Vertical resolution of seismic images is often below the scale of detailed stratigraphic zonation within reservoirs (Sheriff, 1992). This fact implies high uncertainty and low confidence in the position and geometry of the horizons in the structural model built based on the picking of reflectors (Leahy and Skorstad, 2013). The poor quality of seismic data near faults also generate high uncertainty in their interpretation (Thore et al., 2002; Røe et al., 2014). Generally, faults do not generate seismic reflections, and they are

interpreted based on reflector terminations. Nevertheless, reflector terminations can also be produced by other geophysical and geological factors, such as noise in the seismic data, stratigraphic pinchouts, facies changes or unconformities (Alcalde et al., 2017b). Additionally, faults are typically interpreted as singular surfaces whereas they are zones of deformation (Røe et al., 2014).

The time-to-depth conversion is another major source of uncertainty in the structural model, since the true velocity model is

not generally known (Thore et al., 2002; Fomel and Landa, 2014). Average velocities can be estimated from well data, but the lateral coverage is usually low. To amend this, seismic velocities can be used to set up a velocity model for the depth conversion.

Uncertainty associated with wells comes from the well position, well trajectory, and interpretation of well picks (markers, well tops) (Stenerud et al., 2012; Howley and Meyer, 2015; Pakyuz-Charrier et al., 2018). Particularly, capturing well uncertainty

is challenging in reservoirs with multiple horizontal wells, where the adjustment of horizons to honor well information is often in conflict. This uncertainty accumulates when going down the wellbore.

Up to now, different methodologies to capture the uncertainty in the geological structure have been developed, and some have been included as an integrated part of the modeling workflow. Modeling the uncertainty associated with fault shape and location has been addressed by Hollund et al. (2002), Holden et al. (2003), Røe et al. (2014), and Qu et al. (2015), amongst

others. Manzocchi et al. (2008) evaluate the impact on production of the uncertainty on sedimentological characteristics of shallow marine reservoirs combined with the fault uncertainty associated with fault density and permeability. Seiler et al. (2009) presented a reservoir modeling workflow that evaluates the structural uncertainty, where reservoir models use elastic grids that are deformed according to the top and the basal simulated reservoir horizons. Wellmann et al. (2010) develop a methodology for modeling the uncertainty in the geological structure of the subsurface related to the error, bias or imprecision

in input data. Neumann et al. (2012) present a case study in which a workflow that includes a variety of uncertainty sources (e.g. time-to-depth conversion, fluid contacts, and petrophysical properties) is used to estimate oil-in-place volumes. They produce multiple realizations of the depth surfaces based on Bayesian kriging and simulation using Cohiba tool (Abrahamsen, 1993); then 3D grids for each realization are built and the rest of uncertainties are added to the model. Ramos Pinto et al.





(2017) present a methodology for capturing the uncertainty related to seismic data by using seismic attributes to determine zones of high and low quality in the seismic image.

The integrated modeling workflows presented in Leahy and Skorstad (2013), Aarnes et al. (2014) and Howley and Meyer (2015) are capable to capture uncertainties associated to any property and step of the modeling workflow. These workflows

have evolved from methodologies described in MacDonald et al. (2009), Skjervheim et al. (2012), Stenerud et al. (2012), and in turn integrate algorithms for the horizon simulation based on a Bayesian statistical approach by Abrahamsen (1993), Abrahamsen (2005) and Abrahamsen and Benth (2001).

The aim of this article is to capture the structural uncertainty associated with geophysical data in a highly fractured and small reservoir, and use it to evaluate the impact on the predicted gross rock volumes (GRV). Our case study corresponds to the

Lower Cretaceous and Basal Tertiary Group (Oligocene-Miocene) reservoirs of the Lubina and Montanazo mature oil fields. These fields are located in the north-eastern part of the Valencia Trough, offshore Catalonia in the northwestern Mediterranean Sea. The GRV is a main input in the estimation of reserves. Therefore, it is one of the most relevant parameters to evaluate in uncertainty studies, not only during the appraisal and development stages, but also during the whole field life cycle of a reservoir (MacDonald et al., 2009).

We set up two different workflows to model the structural uncertainty: the manually interpreted case, from here on referred to as the manual case, and a constant uncertainty case. They differ in the accuracy with which the uncertainty around the structural interpretation of horizons and faults is captured. We generate multiple scenarios of the structural model and of the 3D reservoir grid, with varying geometries. Additionally, three different scenarios of the oil-water contact (OWC) depth corresponding to the proven (named low case), probable (intermediate case) and the possible cases (high case), were also considered for the

GRV calculations.

This paper has been organized in three main parts. First, an introduction to the geological setting of the oil fields, followed by a description of the dataset is presented. Second, we describe the methods used to model the structural uncertainty and we document the modeling set-up. Finally, we present and discuss the main results obtained.

## 2 Lubina and Montanazo oil fields

### 2.1 Geological setting

The Lubina and Montanazo oil fields are situated 60 km offshore northeast Iberian Peninsula, in the Mediterranean Sea (Fig. 1). The fields are located in the Valencia Trough, a sub-basin that is part of the northwestern Neogene Mediterranean rift system (Roca, 1994; Maillard and Mauffret, 1999; Roca et al., 1999; Granado et al., 2016; Klimowitz et al., 2018; Roma et al., 2018). The Valencia Trough developed under an extensional tectonic regime from the Oligocene to recent times (Roca et

al., 1999) and is characterized by northeast–southwest-oriented horsts and grabens. Prior to the Neogene extension, the area





was affected by a Mesozoic rift, which was followed by a period dominated by compressional processes during the Paleogene (Roca, 1994).

The main reservoirs in the Lubina and Montanazo fields are the Lower Cretaceous rocks and the Basal Tertiary Group (BTG), which are intensely fractured and affected by diagenesis (Fernandez at al., 2015; Fig. 2). The Cretaceous reservoir is made up

of limestones and marls that were affected by subaerial erosion and karstification during the Paleogene uplift. The BTG is upper Oligocene-lower Miocene in age and was deposited unconformably over the Mesozoic. It is composed of breccias and conglomerates of transitional and marine environments, which resulted from weathering and erosion of the Paleogene paleohighs, and of reefal carbonates (Rodríguez-Morillas et al., 2013). During the Miocene rifting, erosion and local karst rejuvenation of the Cretaceous carbonates and the BTG in the structural highs took place.

The Casablanca Group was deposited above the BTG in the early Miocene (Fig. 2). It is mainly composed of black marlstones, which is the main source rock in the area, and hemipelagic marly limestones and mudstones. The source rock contains kerogen type-II and started to generate oil during the Pliocene. The main local seals of the Lubina and Montanazo reservoir are the mudstones and marly limestones of the Casablanca Group and the limestones and mudstones of the middle Miocene San Carlos Group. The mudstones of the upper Miocene Castellon Group is the regional seal in the area (Rodríguez-Morillas, et al., 2013)

(Fig. 2).

The trap of Lubina and Montanazo fields consists in a SSW-NNE trending structural paleohigh, with at least three distinct closures along strike (Fernandez et al., 2015). In detail, the paleohigh is separated in two zones: the Lubina culmination that forms the northeastern part of the structure and the Montanazo culmination, which corresponds to the southwestern part (Fig. 1a). The oil-water contact (OWC) for the Lubina and Montanazo culminations are located at different depths indicating that

the reservoirs in both culminations are disconnected. The Montanazo and Lubina fields produced from Lubina 1 and Montanazo D-5 wells, respectively, since 2009.

## 1.2 Dataset

The study area is 1.1 km wide and 5 km long, and includes both the Lubina and the Montanazo culminations. The dataset was provided by Repsol and consists of:

o  A 3D seismic cube with the interpretation of 6 stratigraphic horizons and 21 faults, which delineate the structure of the reservoir. The horizons include, from top to base: 1) the uppermost horizon (top of the model), which extends throughout the study area and is located between 80 m and 200 m above the underlying stratigraphic horizons; 2) the top of the BTG in the Montanazo culmination; 3) the top of the upper BTG and 4) the top of the lower BTG in the



Lubina culmination; 5) the top of the Cretaceous, which is an unconformity present along the study area; and 6) the basal surface (base of the model), a surface located approximately 200 m below the Cretaceous unconformity.

o Five wells (Lubina 1, Montanazo C-1, Montanazo D-5, Montanazo D-4 and Montanazo D-2; Fig. 1a) with the position of three stratigraphic horizons corresponding to the top of the BTG in the Montanazo culmination, the top of the lower
BTG in the Lubina culmination, and the top of the Cretaceous.

o An interval velocity model and a check shot survey.

## 3 Methods for the structural uncertainty modeling

### 3.1 Manual and constant uncertainty workflows

The sources for uncertainties considered in this work are related to the seismic interpretation of both horizons and faults. These
uncertainties were captured using the structural uncertainty modeling tools in Roxar RMS®, by Emerson Automation Solutions. These tools enable to design a series of processes that are organized in an automated workflow. Thus, the models obtained can be easily revised and updated. A diagram showing the main steps in the uncertainty modeling workflow described in this work is shown in Fig. 3.

Low seismic quality or diffraction areas represent zones of ambiguity in the position of horizons and faults. We captured these
ambiguous zones by defining uncertainty envelopes around an initial horizon and fault interpretation while interpreting (Fig. 4). The definition of the envelopes can be performed either manually while interpreting or automatically as a constant value across the whole surface. In the manual uncertainty case, the uncertainty envelopes are defined by adjusting their dimensions around each interpreted surface in accordance with the variations in the diffractions and in the amplitude of the seismic data. A detailed understanding of the data, and the acquisition and processing of the seismic image are required to properly determine
the constant uncertainty value (Leahy and Skorstad, 2013).

A structural model reproducing the best estimate interpretation is required prior to the uncertainty modeling. In this paper, this is referred as base case model (Fig. 3). It was initially built in time domain using the 21 fault and 6 horizon traces interpreted in the seismic cube. Subsequently, it was converted to depth domain applying the interval velocity model. The base case model (in time or in depth domain, depending on the modeling workflow) was subsequently used to build a structural model
considering the structural uncertainty in geophysical data. The faults and horizons in the base case are references to build new structural models integrating the estimated uncertainty representing realistic structural scenarios.

The uncertainty of faults and horizons can either be considered together or separated in the modeling process. When the fault uncertainty is added a new structural model is created (called New structural model in Fig. 3). Subsequently, the horizon uncertainty is included and a structural model with both the fault and horizon uncertainty is finally obtained (called Final
structural model in Fig. 3). The structural models obtained considering the uncertainties can be combined with other steps in the reservoir modeling workflow (e.g. 3D gridding, property modeling, flow simulation). The effect of structural uncertainties on the volume estimations can be highly non-linear, and the only way to do a proper evaluation is to make stochastic





simulations of the geometries. By integrating structural variations in the overall workflow, the interaction with other parameters can also be evaluated.

The data used in the modeling set-up for both workflows are summarized in Fig. 5. The constant uncertainty case was built from the base case model in depth domain, whereas the manual uncertainty workflow was based on the structural model in time (Fig. 3).

### 3.2 Fault uncertainty modeling

Simulation of the faults is the first step in the proposed workflow (Fig. 3). Different solutions or realizations of fault surfaces considering the related uncertainty are modeled using Monte Carlo simulation (Holden et al., 2003). The simulation results in fault positions changing laterally within the interpreted envelope (Fig. 4). The size of the envelopes or displacement can differ between the hanging wall and the footwall of the faults. In addition, uncertainty in the strike, dip and throw of faults can be evaluated. For the constant uncertainty case, a specific constant envelope dimension for each fault was selected; the values applied ranged between 17.3 m and 35.5 m (Fig. 5). In the interpreted case, this envelope was adjusted according to the seismic quality for each of the faults.

New truncations between faults or removed truncations originally defined in the initial structural model can occur amongst some of the structural model scenarios obtained considering the fault uncertainty, and they need to be checked. The fault changes can also affect the horizon interpretations in zones near the faults. To address this problem, the horizon data in the initial structural model were filtered out around the faults. This is a common procedure to get a realistic fault displacement across the fault plane. Finally, a new structural model with the fault realizations simulated was subsequently built using the filtered horizon data as input data (Fig. 3).

### 3.3 Horizon uncertainty modeling

In the horizon uncertainty modeling, each horizon is modeled as the sum of a trend and a residual (Figs. 6 and 7). The trend is used to reproduce the large-scale variation in the location and shape of the horizons and corresponds to the best estimate interpretation. The residual enables to capture small-scale or local variations between the trend and the unknown true depth of the horizon. It corresponds to differences between data and trends, and allows the simulated trends to be adjusted to match the well observations. The residual is assumed to be a Gaussian random field with a mean of zero and standard deviation as specified by the user (Abrahamsen; 1999). Vertically, the residual is defined by a standard deviation value, whereas its lateral continuity is determined by a variogram.

To capture the total horizon uncertainty, the uncertainty in the trend and the residual are specified (Fig. 7c, d). The values of these uncertainties are associated with many factors, including the quality of seismic data, the uncertainty of the velocity model, the well density and quality of well logs, and the confidence in the interpretations of data. The trend uncertainty is used to constrain how much a trend is allowed to shift vertically, and the residual uncertainty is used to constrain how much a





surface can deviate away from the trend to adjust to the well data. The variogram for the residual will determine the shape of the noise when simulating surfaces.

The horizons can be modeled deterministically or stochastically (i.e. Monte Carlo simulation). The deterministic or predicted method represents the average results of all the simulations. In prediction mode, the trend is shifted according to a posterior trend coefficient, which is calculated from the prior uncertainty and the well observations (Abrahamsen et al., 2018) (Fig. 7e). To fit the well data, the trend is locally adjusted to the well picks by kriging within the residual uncertainty, and a new horizon model (only one solution) is built (Fig. 7f). With this method, a prediction error or local depth uncertainty maps are obtained based on Bayesian updates of the uncertainty envelopes. The prediction error is minimum in the well locations, since the trend honor the well data, and increases away from the wells following the variogram (Fig. 8).

The stochastic simulation produces multiple, equi-probable realizations of the horizons, and spans the uncertainty. The trend coefficients with which the trends are shifted are drawn from the posterior uncertainty distribution (Fig. 7g). Subsequently, the simulated residual field is adjusted to the wells with Bayesian kriging and added to the trend (Fig. 7h).

In the manual uncertainty workflow, the vertical residual uncertainty was defined using standard deviation maps extracted from the uncertainty envelope interpreted in time domain (Fig. 5). Standard deviation values in the maps ranged from 3.23 ms to 5.49 ms. In the constant uncertainty case, a constant standard deviation for each horizon was applied for the vertical residual uncertainty. Values between 4.2 m and 6.54 m, depending on the horizon, were used (Fig. 5). These values were estimated by obtaining the standard deviation of the averaged differences between the best estimate interpretation and the immediate zero amplitudes of reflectors in a number of sampled locations. For the spatial continuity of the residual uncertainty, a horizontal variogram range of 500 m was set for all the horizons to fit all data and a spherical variogram function was selected. In both workflows, a value of 0.3% of the trend was applied to set the trend uncertainty for all the horizons.

For the prediction method, the fault uncertainty was not modeled and only one solution of the structural model with the horizon uncertainty was built. For the simulation method, the horizon uncertainty modeling process was added to the structural model that captures the fault uncertainty to generate one horizon model for each realization of the fault model. In this way, one different structural model for each realization was obtained (i.e. final structural model in Fig. 3). We simulated 200 realizations of the structural model of each uncertainty modeling workflow. For the manual uncertainty case, which was based on the base case model in time, the horizons and faults in each realization were converted to depth domain with the interval velocity model to obtain a final structural model. For the constant case, the uncertainty modeling workflow was entirely run in depth domain.

## 3.4 Three-dimensional grid and GRV

For the base case structural model, for the prediction models and for each realization simulated, a 3D grid was built. The corner point gridding method along with pillar gridding for faults (i.e. planar surfaces) was selected. In plan view, cell dimensions were set to 25 m by 25 m, and vertically, the cell height was set to 20 m. This resulted in 3D grids with 1.5 million cells and approximately half million defined cells. Each 3D grid was quality controlled to verify the lack of collapsed cells.





Finally, the gross rock volume of the reservoir (i.e. Lower Cretaceous and BTG) above the OWC was estimated for the base case model, the prediction models and for each realization considering three scenarios: a) the low case (proven case), where the OWC is at 2344 m and at 2451 m below sea level in the Montanazo and Lubina culminations, respectively; b) the intermediate case (probable case), with contacts at depths of 2360 m and 2458 m; and c) the high case (possible case), with contacts at 2372 m and 2465 m, respectively.

## 4 Results

### 4.1 Base case model

The base case model of the Lubina and Montanazo fields shows a SW-NE oriented structural high, bounded by two major normal faults to the NW and SE (Fig. 9). Internally, other minor normal faults with a lower displacement and variable orientation compartmentalize the reservoir in multiple blocks. In the center of the area, a NW-SE oriented fault separate the Lubina and Montanazo culminations.

The Montanazo culmination is approximately 2.4 km long and 0.8 km wide, and the Lubina culmination is 2.2 km long and 0.6 km wide. The BTG reservoir shows significant thickness changes throughout the field. In the Lubina culmination, both the upper and lower BTG reservoirs thins out towards the SSW, and the upper BTG pinches out few meters to the SSW of the Lubina-1 well position (Fig. 9c).

The maximum vertical thickness of the reservoir in the Montanazo culmination between the top of BTG Montanazo horizon and the OWC is 109 m in the low case, 125 m in the intermediate case and 137 m in the high case. In the Lubina culmination, the maximum vertical thickness recorded between the top of the upper BTG Lubina (i.e. top of the reservoir in this culmination) and the OWC is 133 m, 140 m and 147 m for the low, intermediate and high cases, respectively. Values of 80.5 x $10^6$ m$^3$, 101.6 x $10^6$ m$^3$ and 122.6 x $10^6$ m$^3$ for the GRV above the OWC at the low, intermediate and high cases were estimated from the base case model (Figs. 10a).

### 4.2 GRV in the prediction models and in the simulated realizations

Changes in geometry, structure and dimensions of the reservoir in the Lubina and Montanazo culminations with respect to the base case model have been captured in the predictions models and in the 200 realizations of the manual and the constant uncertainty workflows. These variations impact on the GRV estimated, as it is documented in Figs. 10 and 11. Figure 10a summarizes the results of the volume calculations for the prediction models and for realizations corresponding to the proved (i.e. P10), the probable (i.e. P50) and the possible (i.e. P90) scenarios, as well as complementary statistics. The GRV distribution for the 200 realizations obtained in the manual and in the constant cases for the three scenarios of the OWC depth are documented in the histograms of Fig. 10b and c and in the box plots of Fig. 11 that also shows the volumes for the base case and prediction models. In Fig. 12, the GRV distributions in the 3D grids for a selection of models are presented.



Figures 13, 14 and 15 show three cross sections of the structural models for the manual and the constant uncertainty workflows. The realizations presented in the cross sections are those corresponding to P90, P50 and P10 of the GRV for the intermediate case of OWC depth (i.e. OWC at 2360 m and 2458 m below sea level for the Montanazo and Lubina culminations, respectively). The cross section B-B' (Fig. 13) is a transect throughout the Montanazo culmination, whereas the cross sections

C-C' (Fig. 14) and D-D' (Fig. 15) show the structure of the Lubina culmination in a southern and in a northern positions, respectively.

The GRV estimated for the prediction model in the manual uncertainty case are $81.8 \times 10^6$ m$^3$ for the low case, $103.2 \times 10^6$ m$^3$ for the intermediate case, and $124.5 \times 10^6$ m$^3$ for the high case (Fig. 10a and 11). For the prediction model in the constant case, the estimated GRV correspond to $81.6 \times 10^6$ m$^3$, $102.8 \times 10^6$ m$^3$, and $123.8 \times 10^6$ m$^3$ for the three OWC scenarios, respectively.

The mean GRV of the 200 realizations in the manual uncertainty case are $80.4 \times 10^6$ m$^3$ for the low case, $101.6 \times 10^6$ m$^3$ for the intermediate case and $122.7 \times 10^6$ m$^3$ for the high case. For the constant uncertainty case, the mean GRV are slightly lower; $77.7 \times 10^6$ m$^3$ for the low case, $98.4 \times 10^6$ m$^3$ for the intermediate case and $118.9 \times 10^6$ m$^3$ for the high case. The median for all simulation cases are very close to the respective mean value, with maximum differences of $0.5 \times 10^6$ m$^3$ (Fig. 10b, c). The ranges (i.e. difference between the maximum and the minimum values) of the GRV in all realizations for the manual and the

constant uncertainties are, respectively, $30.4 \times 10^6$ m$^3$ and $33.4 \times 10^6$ m$^3$ for the low case, $33.1 \times 10^6$ m$^3$ and $36.2 \times 10^6$ m$^3$ for the intermediate case, and $35.4 \times 10^6$ m$^3$ and $39.1 \times 10^6$ m$^3$ for the high case (Figs. 10a and 11).

In the manual uncertainty case, a difference of $12 \times 10^6$ m$^3$, $14 \times 10^6$ m$^3$ and $14 \times 10^6$ m$^3$ of GRV between the realizations corresponding to P10 and P90 scenarios are recorded, respectively, for the lower, intermediate and higher OWC depths. In the constant uncertainty case, the difference increases up to $15 \times 10^6$ m$^3$, $17 \times 10^6$ m$^3$ and $20 \times 10^6$ m$^3$ for the three OWC scenarios

(Fig. 10).

## 5 Discussion

### 5.1 Manually interpreted versus constant uncertainty cases

The prediction method yielded similar responses in the GRV from both workflows, with GRV between 1% and 1.6% higher than the base case in all the OWC depth scenarios (Fig. 11). The relatively low impact on the GRV predicted is attributed to

the fact that the prediction method applied only captures the horizon uncertainty, not the fault uncertainty, and only the most likely horizon position is modeled, which may blur the difference between the workflows compared.

The GRV values obtained from the 200 realizations of each of the modeling scenarios considered show a distribution that is close-to-normal, as indicated by mean values very close to the median, and the bell shape of the calculated histograms (Fig. 10). Thus, the predicted GRV in each scenario can be considered as representative of the variability that the structural

uncertainty estimation can generate with regard to an initial best estimate structure.

Significant differences between the manual and the constant uncertainty workflows were recorded in the GRV distributions of the simulated realizations (Figs. 10 and 11):



- o The mean and median GRV for the constant uncertainty case are below the GRV in the base case model for the three scenarios of the OWC depth. By contrast, in the manual uncertainty case, the mean and median are higher than the base case GRV but very close to it.

- o The P10, P90, minimum and maximum GRV values (in addition to the median) are lower in the constant than in the manual uncertainty case. So, in general, lower GRV values are recorded for the constant than for the manual uncertainty cases.

- o The ranges (i.e. difference between maximum and minimum values) in the constant uncertainty case are systematically higher (between 3.7 and 3.1 x $10^6$ m$^3$) than in the manual uncertainty case. This indicates a higher dispersion of the results in the former case.

The same uncertainty trend value and kriging method for the trend simulation, and the same variogram range for the horizontal residual uncertainty were applied in both the manual and the constant uncertainty workflows. Hence, the differences in the GRV observed in both cases is expected to result from the envelopes around the horizon and fault interpretations.

Finding the right level for the residual uncertainties can be a challenging task that requires many iterations and detailed knowledge of the data. As the choice of residual uncertainty has large impact on the simulated horizons, this choice will also affect the GRV estimates. However, by capturing the uncertainty as envelopes while interpreting, we get an accurate and consistent representation of the uncertainties directly from the seismic data.

Differences attributed to the type of uncertainty envelopes defined are also recorded in the uncertainty maps describing the prediction error obtained from the prediction models (Fig. 8). In both the manual and the constant cases, the prediction errors are low in the well locations for those horizons with well data available, as expected.

## 5.2 Impact of the structural uncertainty in a small and fractured reservoir

Our study focuses on a small and highly fractured reservoir, which exhibits significant thickness changes and irregular horizon geometries (Figs. 9 and 13 to 15). GRV predicted from the realizations in the manual and constant uncertainty cases estimate significant variations in GRV with respect to those calculated from the base case model (Fig. 11 and 12). In terms of percentage difference, the realizations record mean values of 16% above and below the GRV in the base case for the manual case considering the three OWC depth scenarios. Mean differences in the constant case are 22% lower and 14% higher the GRV in base case model.

As shown in Fig. 12a – e, for the same case of OWC depth (i.e. Intermediate depth) and considering the realization corresponding to P50 of the GRV in the stochastic realizations, variable extent of the oil zone is predicted amongst the base case, the prediction models and the simulated models. This is more prominent when considering realizations corresponding to the possible (P10) and proven (P90) scenarios. The realization corresponding to the proven volume in the manual case (Fig. 12g) shows that the Lubina culmination is not fully saturated in oil, since the central part is below the OWC in the intermediate depth, by contrast with the possible case in which almost the entire Lubina culmination is above the OWC (Fig, 12f). In one of the most optimistic scenarios represented by the possible case (P10) and the high OWC depth scenario (Fig. 12h), the oil




zone covers almost the entire Lubina culmination (similarly to the realization P10 for the intermediate case; Fig. 12f) and, additionally, the oil zone in the Montanazo culmination widens towards the southeast. In the scenario corresponding to the proven volume (P90) and the low OWC depth case (Fig. 12i), the oil zone is significantly reduced.

The differences in the GRV and in the oil zone distribution that have been recorded by the uncertainty modeling workflows

indicate that including the structural uncertainty associated with the picking of horizons and faults can be relevant to predictions derived from the structural model, even for small reservoirs with a relatively well constrained structure.

The differences in the GRV can be, to some degree, correlated with variations in the reservoir structure (Figs. 13 to 15). It can be noted that the distance between the major NE-SW oriented faults that laterally limit the reservoir (see Fig. 9a, b) increases from realization P90 (proven case) to realization P10 (probable case) (Figs. 13c, 14c and 15c). Therefore, the greater separation

of the major faults owing to their lateral displacement within the fault uncertainty envelope leads to increase the width of the reservoir with the result of enhancing the predicted GRV. This fact indicates that the fault uncertainty in major faults strongly impacts the GRV in the reservoir.

By contrast, no systematic variation in the positions of the horizons (i.e. decreasing or increasing depth) has been observed comparing the P10, P50 and P90 realizations in Figs. 13d, 14d and 15d. This is as expected, as the residual Gaussian field has

an expectation of zero, and the uncertainty of the trend is relatively low.

### 5.3 Uncertainty modeling workflows

We have modeled the structural uncertainty using modern workflows, whose main advantages and limitations encountered in this study are discussed below.

The application of these workflows generate multiple realizations of the structural model that are all geologically consistent

with the available data. Moreover, all realizations can be quickly updated when required, as for example when new fault and horizon interpretations are performed, when input data is changed, or new well data is available.

The workflows consist of a chain of processes that can be easily designed. However, a previous knowledge of the data is necessary to set up the uncertainty modeling appropriately. The uncertainty trend should be defined according to the distance the horizons can be moved away from the original interpretation. Trend uncertainty values are typically up to 10% of the depth

of the horizon, but this can vary depending on the field and the amount of data available. As modeling the horizon uncertainty is directly related to the resolution of the seismic data, the trend uncertainty generally increases with depth (Fig. 16). The uncertainty drops to zero in the well locations (since the simulated horizons are adjusted to match the well picks) and gradually increases away from the wells according to the variogram settings (Fig. 16). Therefore, in reservoirs with few well data is especially relevant to capture the structural uncertainty.

The variogram ranges set for the residual uncertainty should in general not be more than the half of the reservoir size, and represent the expected lateral continuity of the geological layering. In this study, the variogram range was set as to the half of the reservoir width (i.e. 500 m). The variogram function should be selected according to the degree of noise to be captured by the residual. Exponential functions are used for erratic scenarios, Gaussian variograms produce smooth results, and spherical




functions (the variogram type used in this study) are in between both the exponential and Gaussian functions and tend to represent realistic scenarios in geosciences.

As documented and discussed above, manual interpretation of the uncertainty envelopes around faults and horizons will produce more accurate predictions of the uncertainty impact. This method may appear more time-consuming than the constant uncertainty case, but finding the appropriate constants can be equally challenging and time consuming, if not even more.

Adding the structural uncertainty in the modeling workflow has allowed us to evaluate its effects on the reservoir volume estimation in this case study. In further studies, the impact of the structural uncertainty can also be assessed on a variety of outcomes of the reservoir modeling process, such as facies proportions, recovery efficiency and water cut predictions, and in history matching. Additionally, other sources of the structural uncertainty, such as the velocity uncertainty and well uncertainty, can also be investigated. Moreover, the structural uncertainty can be combined in the modeling workflow with the uncertainty associated with other parameters, such as porosity, fluid contacts, fluid saturations, fault seal or aquifer size to produce a set of possible scenarios whose variations account for all the uncertainties considered.

## 6 Conclusions

We have applied modern workflows to document the impact of the geophysical uncertainty on predicted GRV in the small and highly fractured Lower Cretaceous and Tertiary reservoir of the Lubina and Montanazo fields (western Mediterranean Sea). The results obtained allowed us to conclude that:

- o The reservoir exhibits significant thickness changes and irregular horizon geometries. It is laterally limited by major NE-SW oriented normal faults, and internally, is compartmentalized in multiple blocks by other minor faults. The Montanazo and Lubina culminations are separated by a NW-SE oriented normal fault.
- o The uncertainty modeling workflows applied generate multiple realizations of the structural model showing differences in geometry and dimensions of the reservoir, but which are all geologically consistent with the available data. Capturing the structural uncertainty by producing multiple realizations allows us to improve the understanding of the reservoir structure.
- o The structural uncertainty associated with the picking of horizons and faults in seismic data has a relevant impact on the volume estimation. The realizations capturing the structural uncertainty predict mean GRV percentage differences with respect to the GRV in the base case (the model from the best estimate interpretation) that are up to 16% higher and 22% lower.
- o The two uncertainty modeling workflows tested in this study differ in the accuracy with which the uncertainty around the horizons and fault interpretations in the seismic data is defined. Capturing the uncertainty by defining manually the ambiguous zones around the interpreted surfaces in the seismic data reports narrower GRV predictions from the



simulated structural models, more accurate horizon prediction errors and more consistent results than modeling the structural uncertainty with constant sizes for the ambiguous zones.

o The major NE-SW oriented normal faults that bound laterally the reservoir impact to a larger extent the GRV predicted from the simulated structural models.

o Uncertainty in the horizons increases with depth (trend uncertainty) and with the absence of well picks available. Therefore, for those scenarios with few well data available capturing the structural uncertainty is especially relevant.

o The uncertainty modeling workflows applied are easy to design and allow to update the models when required. As always, domain knowledge of the data is necessary to set up appropriately the uncertainty modeling workflows.

o In next steps of the modeling, it may be significant to use a single workflow combining other sources of structural

uncertainty (i.e. velocity uncertainty and well uncertainty) to obtain a more complete estimation of the structural uncertainty in the GRV. Covering the entire range of possible structural scenarios will help to understand better the geology of Lubina and Montanazo oil fields by adding more value to the simulation and history matching stages. This will lead to an improved field development decision making process. Furthermore, the grids constructed for each realization could be the base to build property models and further simulation, allowing all uncertainties to be preserved

and thus, the risk assessment to be improved.

**Author contribution**

Alexandre Bouche and Ingrid Aarnes were responsible of the conceptualization of the research, and Carlos Gordillo of providing the original data set and interpretation by Repsol. Carla P. Bárbara and Patricia Cabello conducted the research, and along with the rest of co-authors performed the formal analysis of the results. Patricia Cabello, Carla P. Bárbara and Ingrid

Aarnes prepared the original manuscript which was reviewed and edited with contributions from all co-authors.

**Competing interests**

The authors declare that they have no conflict of interest.

**Acknowledgements**

Thanks are due to Emerson Automation Solutions for donation of RMS licenses and for the technical support to use the

software. We are indebted to Repsol for providing the data set used and also for the support received. Support from SALCONBELT Project CGL2017-85532-P) and SEROS Project (CGL2014-55900-P), Geomodels Research Institute and Grup de Geodinàmica i Anàlisi de Conques (2017SGR596) is gratefully acknowledged.




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



**Figures**

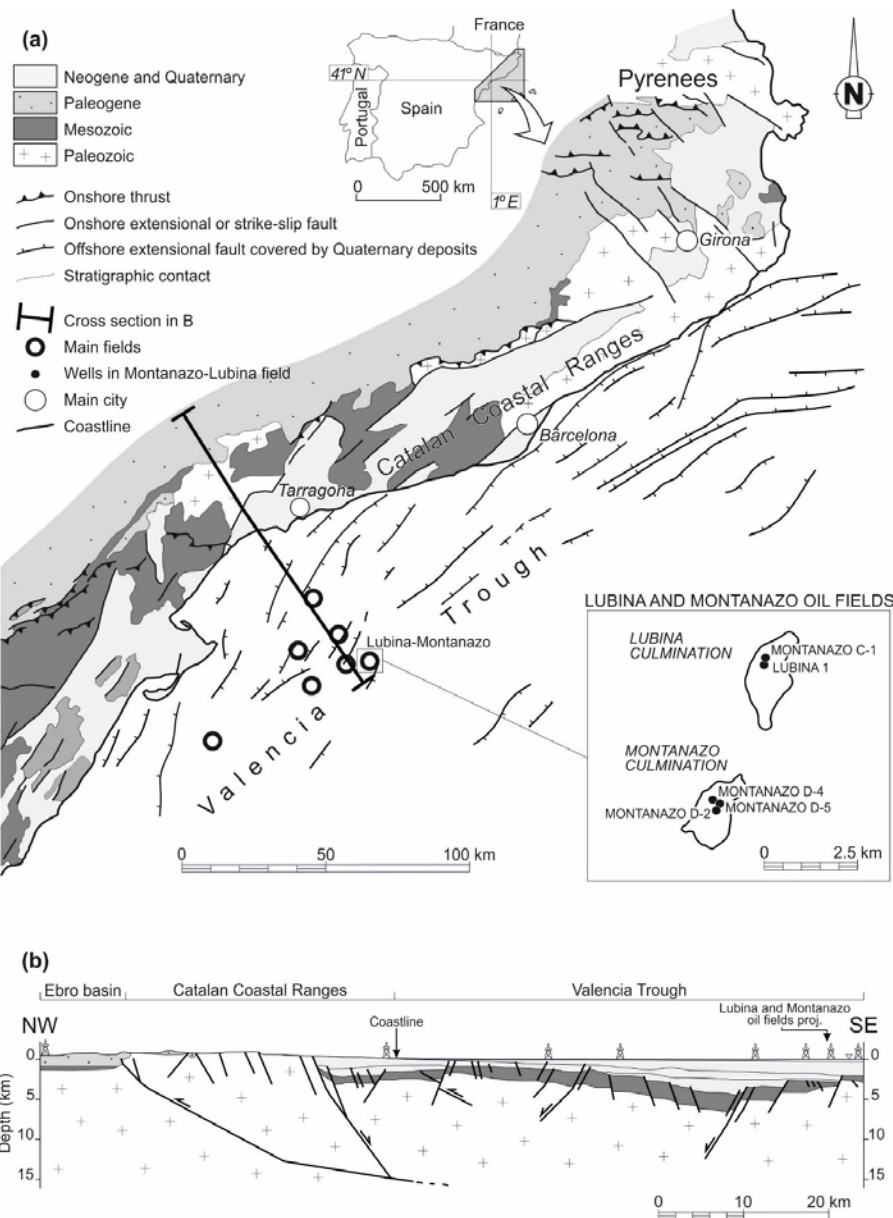

**Figure 1: (a) Geological map of the northeastern part of the Iberian Peninsula showing the location of the Lubina and Montanazo oil fields. The Montanazo culmination forms the southwestern part of the field where Montanazo D-2, Montanazo D-4 and Montanazo D-5 wells were drilled. The Lubina culmination corresponds to the northeastern part and has the Montanazo C-1 and Lubina 1 wells. Modified from Roca (1994) and Roca et al. (1999). (b) Regional cross section from the Ebro basin to the Valencia Trough located few kilometers south of Lubina and Montanazo oil fields. Modified from Cabrera et al. (2004).**



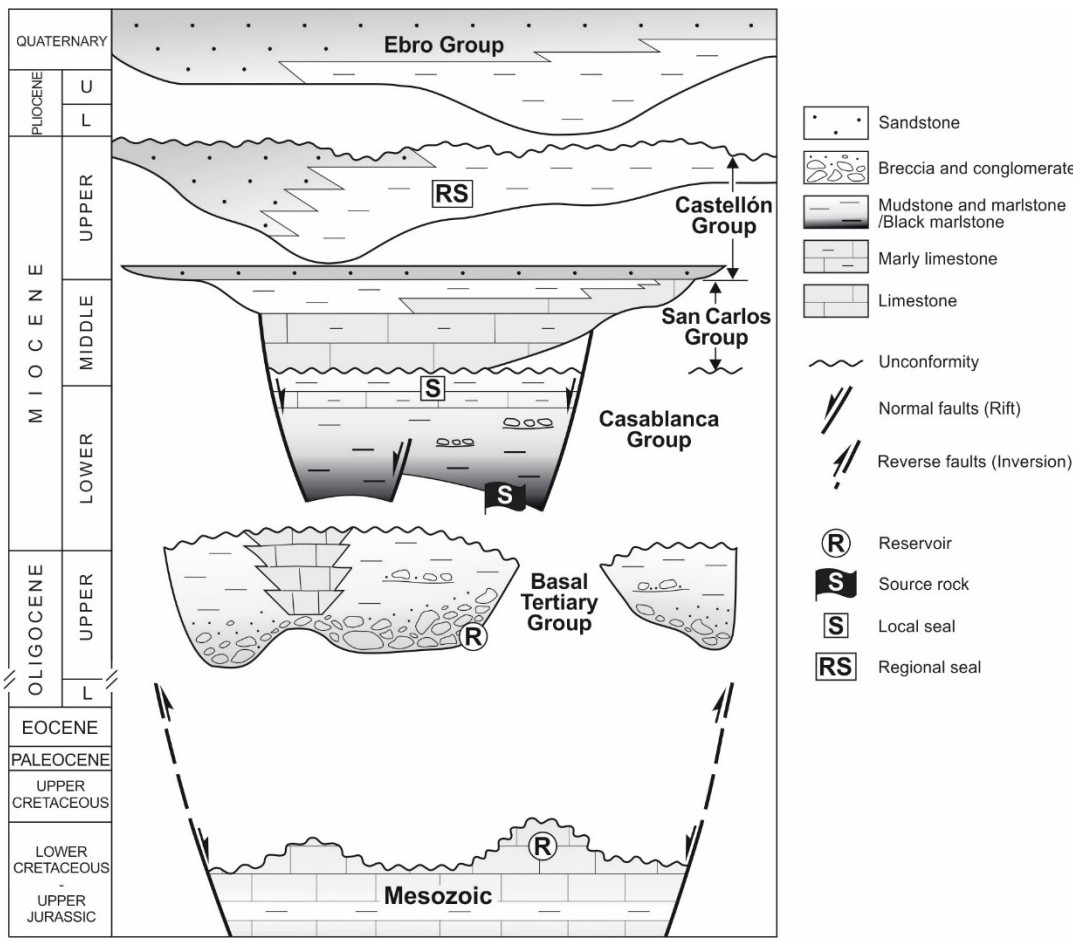

**Figure 2: Chronostratigraphic diagram of the study area showing the age and main lithologies of the infill of the Valencia Trough in the Montanazo and Lubina field area. The reservoir, source and seal rocks are also indicated.**



**Figure 3: Diagram showing the steps of the workflow applied for both the manual and the constant uncertainty modeling cases. An initial structural model is constructed using the best estimate interpretation of the seismic data. The fault uncertainty, first, and the horizon uncertainty later, are added into the model obtain a final structural model reproducing different scenarios of fault and horizon geometry and position. The scenarios are finally compared in terms of gross rock volumes (GRV) to evaluate the impact of the structural uncertainty. The number of figures illustrating the modeling steps are indicated.**

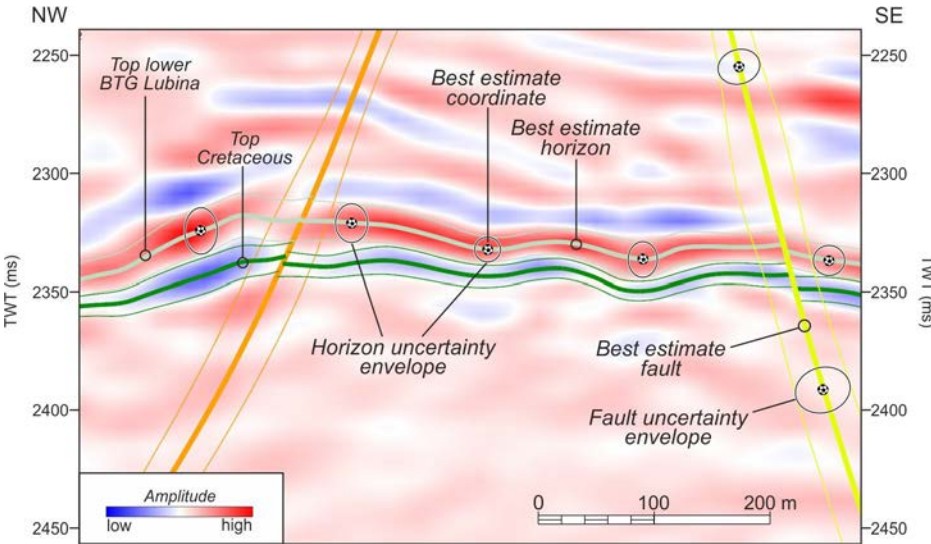

**Figure 4: Best estimate interpretation of the horizons corresponding to the top of the lower BTG in the Lubina culmination and the top of the Cretaceous, and of two faults, and the uncertainty envelope around the best estimate. This example illustrates the manual uncertainty case; note that the envelope sizes vary along the interpretation according to degree of ambiguity in the seismic data.**



| | | | MANUAL UNCERTAINTY TREND | CONSTANT UNCERTAINTY TREND |
|---|---|---|---|---|
| HORIZON UNCERTAINTY MODELING | TREND | Uncertainty trend | 0.33% | 0.33% |
| | | Modeling method | Bayesian kriging | Bayesian kriging |
| | RESIDUAL | Vertical residual uncertainty | Standard deviation maps extracted from envelopes in the seismic interpretation in time | Constant values for each horizon — Top Model 5.3 m · Top Upper BTG Lubina 4.2 m · Top Lower BTG Lubina 4.4 m · Top BTG Montanazo 5.8 m · Top Cretaceous 6.5 m · Base Model 5.8 m |
| | | Horizontal residual uncertainty | Variogram range = 500 m<br>Spherical variogram function | Variogram range = 500 m<br>Spherical variogram function |
| FAULT UNCERTAINTY MODELING | | | Variable distances. Extracted from envelopes in the seismic interpretation in time | Same values for footwalls and hanging walls — Constant values for each horizon — Minimum radius 17.3 m · Maximum radius 36.5 m · Mean radius 28.2 m · Median radius 28.8 m |

**Figure 5: Parameters, values and methods used in the manual and constant uncertainty cases to capture the horizon and the fault uncertainties.**



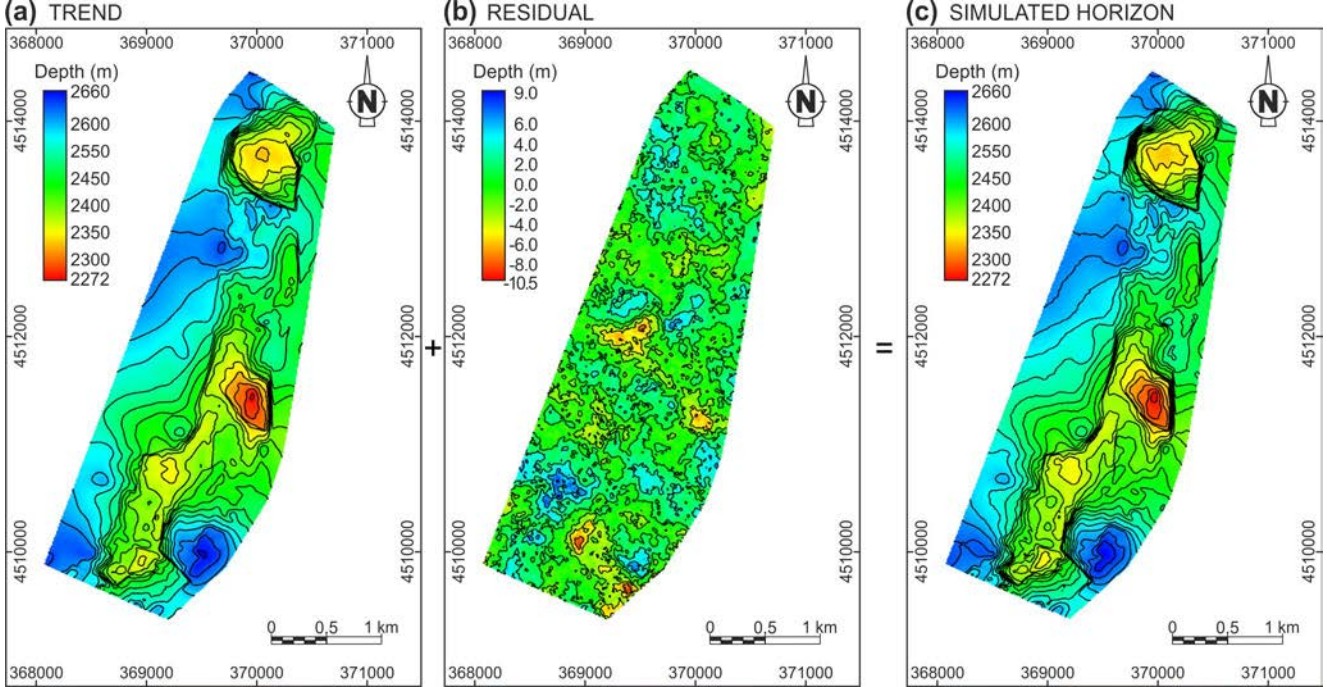

**Figure 6: (a) Trend capturing the large-scale variation in the location and shape of the horizon. (b) Residual reproducing the small-scale or local variations between the trend and the unknown true depth of the horizon. (c) Simulated horizon that results from the sum of the trend and the residual. The example corresponds to the top of the Cretaceous. See also Fig. 7 for more details.**





BASE CASE MODEL

**(a)**

TREND
*Best estimate
horizon*

Well picks

LOCAL WELL ADJUSTMENT

**(b)**

*Best estimate
horizon*

Local adjustment
to wells

HORIZON UNCERTAINTY MODELING

**(c)**

TREND
*Best estimate
horizon*

TREND
UNCERTAINTY

**(d)**

TREND
*Best estimate
horizon*

RESIDUAL
UNCERTAINTY

PREDICTION

**(e)**

PREDICTED
TREND

**(f)**

PREDICTED
HORIZON

SIMULATION

**(g)**

SIMULATED
TREND

Gaussian distribution

**(h)**

SIMULATED
HORIZON

**(i)**

$-3\sigma$
$-2\sigma$
$-\sigma$
TREND — *Best estimate*
$-\sigma$
$-2\sigma$
$-3\sigma$

Range of trend
uncertainty

68% of the realizations
within this interval

95% of the realizations
within this interval

99% of the realizations
within this interval





**Figure 7: Diagram showing the steps of the horizons uncertainty modeling. (a) Initial situation with the best estimate interpretation of a horizon, which corresponds to the trend. This interpretation does not fit the well picks (i.e. wells observations, markers, well tops). (b) If the horizon uncertainty is not considered, typically the horizon interpretation is locally adjusted to the well picks. If the uncertainty is modeled, then the trend uncertainty range (c) and the residual uncertainty (d) are estimated. In the prediction method,**
5    **the most like position of the trend is found (e) and the residual uncertainty is added to locally adjust the trend to the well picks (f). In simulation, a trend is simulated considering the trend uncertainty range and assuming a Gaussian distribution (g). Subsequently, the residual is included to add local variability and to adjust the horizon to the well observations (h). Since a Gaussian distribution is considered for modeling the trend, the estimation of the trend uncertainty range must consider the percentage of the realizations to be modelled within the selected range (i).**



**Figure 8: Horizon uncertainty maps obtained from the prediction method, describing the prediction error for all the horizons in (a) the manual and (b) the constant uncertainty cases. The top of the lower BTG in Lubina, the top of the BTG in Montanazo and the top of the Cretaceous have well picks available (indicated by crosses).**



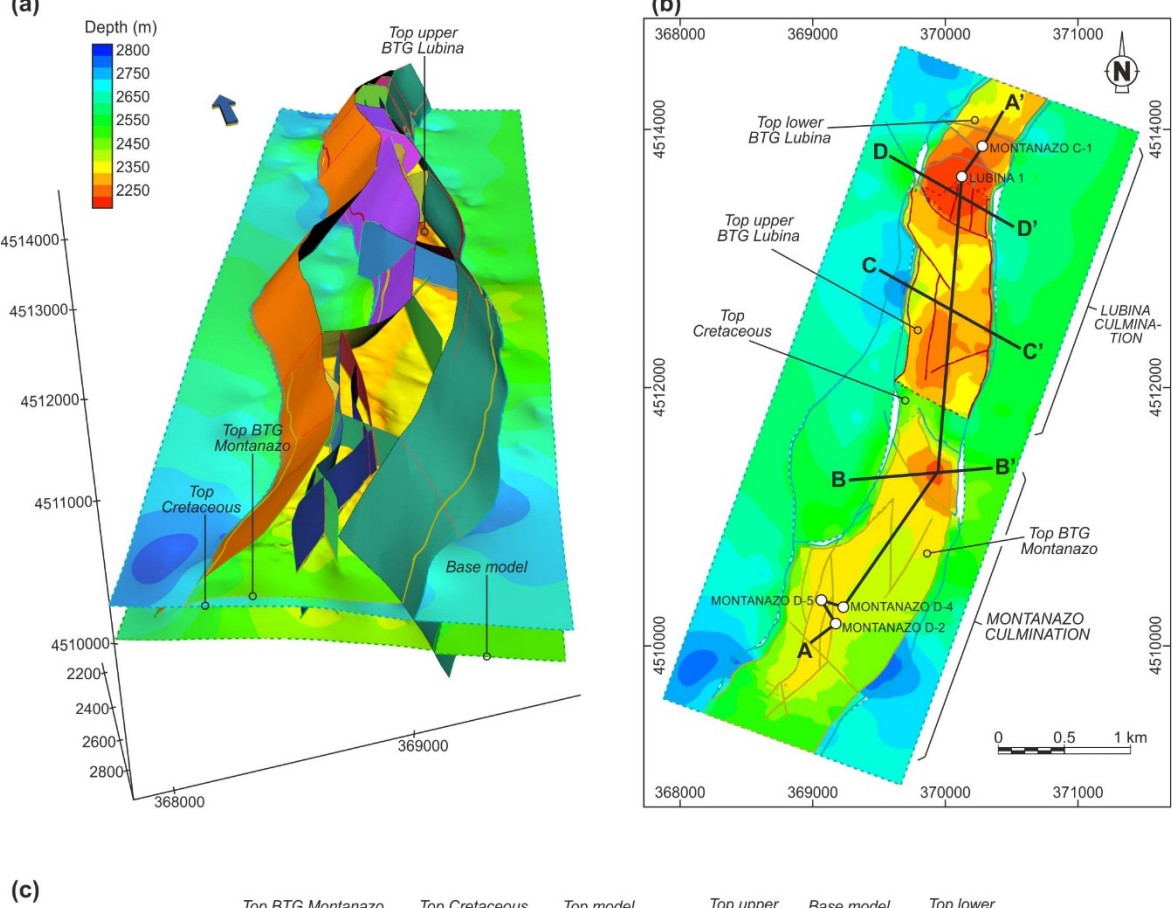

**Figure 9: Base case structural model. (a) Three-dimensional view of the model showing the faults and the horizons reconstructed (with the exception of the uppermost Top model horizon). (b) Map view of the model showing the horizons bounding the reservoir to the top. The location of cross sections A-A', B-B', C-C' and D-D' presented in frame (c) and in Figs. 13, 14 and 15, respectively, are shown. The position of the wells are also indicated. (c) Cross section A-A' of the Base case model showing the structure of the reservoir. Note that the Lubina and Montanazo culminations are disconnected.**




**Figure 10: Gross rock volumes (GRV) calculated from the models obtained with the manual and the constant uncertainty workflows. (a) GRV of the base case model, the predicted models and of realizations corresponding to the proved (P90), probable (P50) and possible (P10) for the manual and the constant uncertainty cases in the low, intermediate and high oil-water contact depth cases. The minimum, maximum, mean and median for all simulation cases are also reported. Histograms showing the GRV distribution obtained from the 200 realizations (simulation method) of (b) the manual uncertainty case and of (c) the constant uncertainty case for the three oil-water contact depth cases. The mean GRV in all cases are also indicated.**





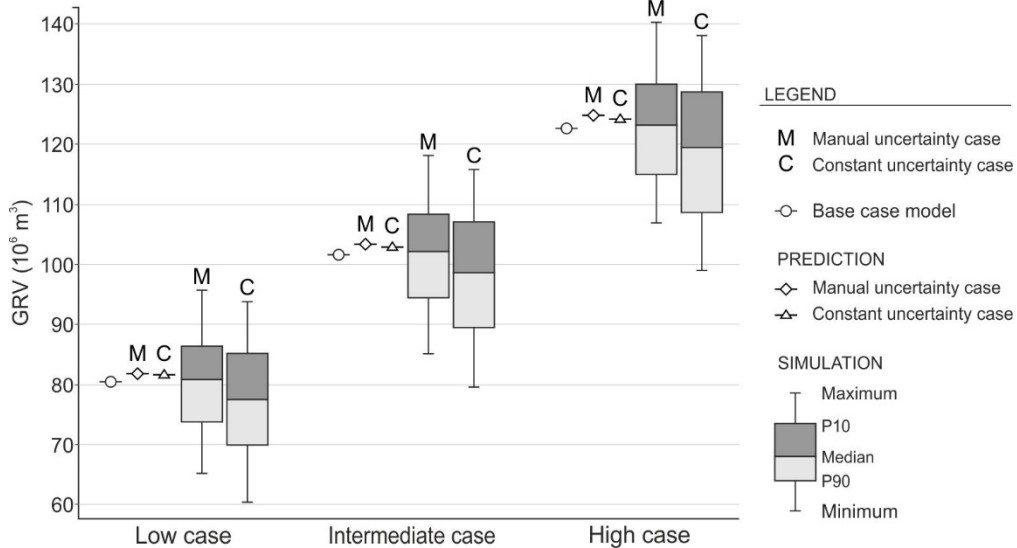

**Figure 11: Box plots of the GRV obtained from the 200 realizations (simulation method) of the manual and the constant uncertainty cases for the low, intermediate and high oil-water contact depths. The GRV of the base case and prediction models are also indicated.**



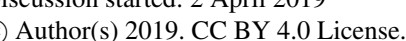





**Figure 12: Top view of the 3D grids with the oil gross volume distribution for: (a) the base case model; the prediction method for (b) the manual and (c) the constant uncertainty case; the P50 realization of (d) the manual and (e) the constant uncertainty cases; (f) the P10 (possible volume) and (g) the P90 (proven volume) for the manual uncertainty case, in the intermediate OWC depth scenario. (h) The P10 in the high OWC scenario, and (i) the P90 in the low OWC scenario for the manual uncertainty case.**




**Figure 13: Cross section B-B' of the structural model at the Montanazo culmination for the intermediate oil-water contact depth case showing the realizations corresponding to the proved (P90), probable (P50) and possible (P10) scenarios of the manual (a) and the constant (b) uncertainty cases. (c) Diagrams showing the position of the intersection between the major faults bounding laterally the reservoir and the oil-water contact for the P90, P50 and P10 scenarios. (d) Diagrams showing the maximum vertical distance between the reservoir top and the oil-water contact for the P90, P50 and P10 scenarios. See the position of the section in Fig. 9b.**



**Figure 14: Cross section C-C' of the structural model in the southern sector of the Lubina culmination for the intermediate oil-water contact depth case showing the realizations corresponding to the proved (P90), probable (P50) and possible (P10) scenarios of the manual (a) and the constant (b) uncertainty cases. (c) Diagrams showing the position of the intersection between the major faults bounding laterally the reservoir and the oil-water contact for the P90, P50 and P10 scenarios. (d) Diagrams showing the maximum vertical distance between the reservoir top and the oil-water contact for the P90, P50 and P10 scenarios. See the position of the section in Fig. 9b.**



**Figure 15: Cross section D-D' of the structural model in the northern sector of the Lubina culmination for the intermediate oil-water contact depth case showing the realizations corresponding to the proved (P90), probable (P50) and possible (P10) scenarios of the manual (a) and the constant (b) uncertainty cases. (c) Diagrams showing the position of the intersection between the major faults bounding laterally the reservoir and the oil-water contact for the P90, P50 and P10 scenarios. (d) Diagrams showing the maximum vertical distance between the reservoir top and the oil-water contact for the P90, P50 and P10 scenarios. See the position of the section in Fig. 9b.**



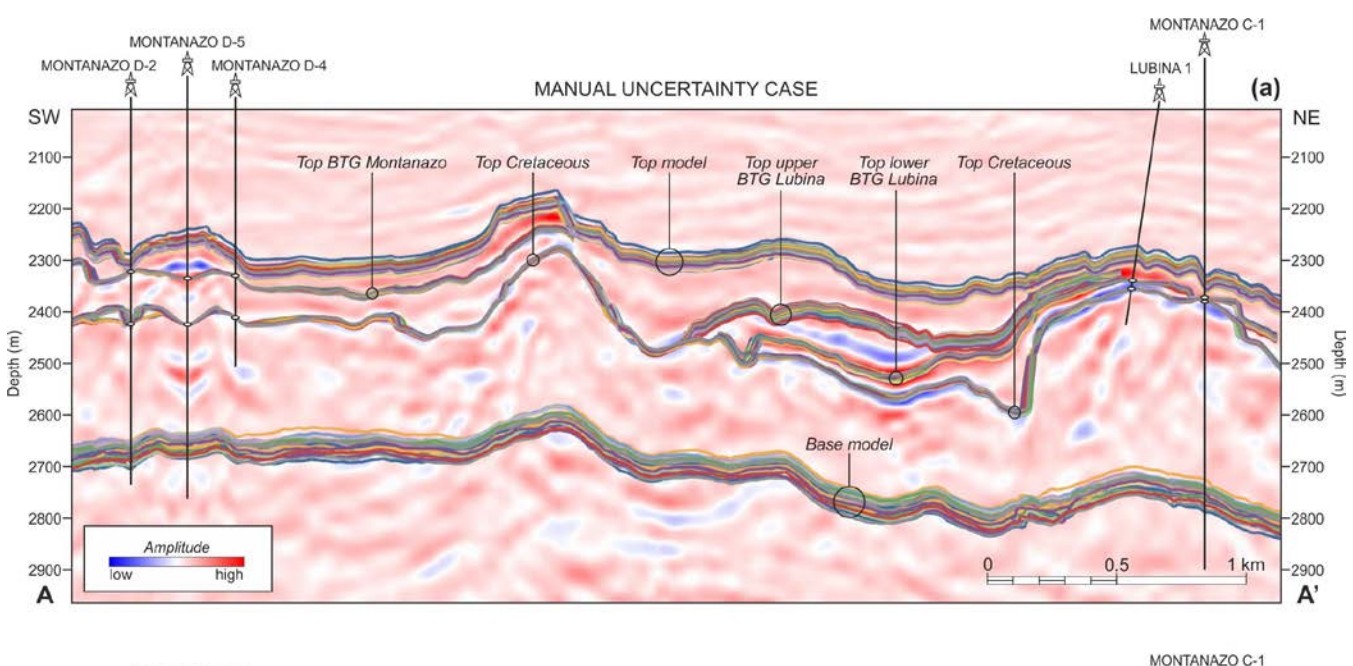

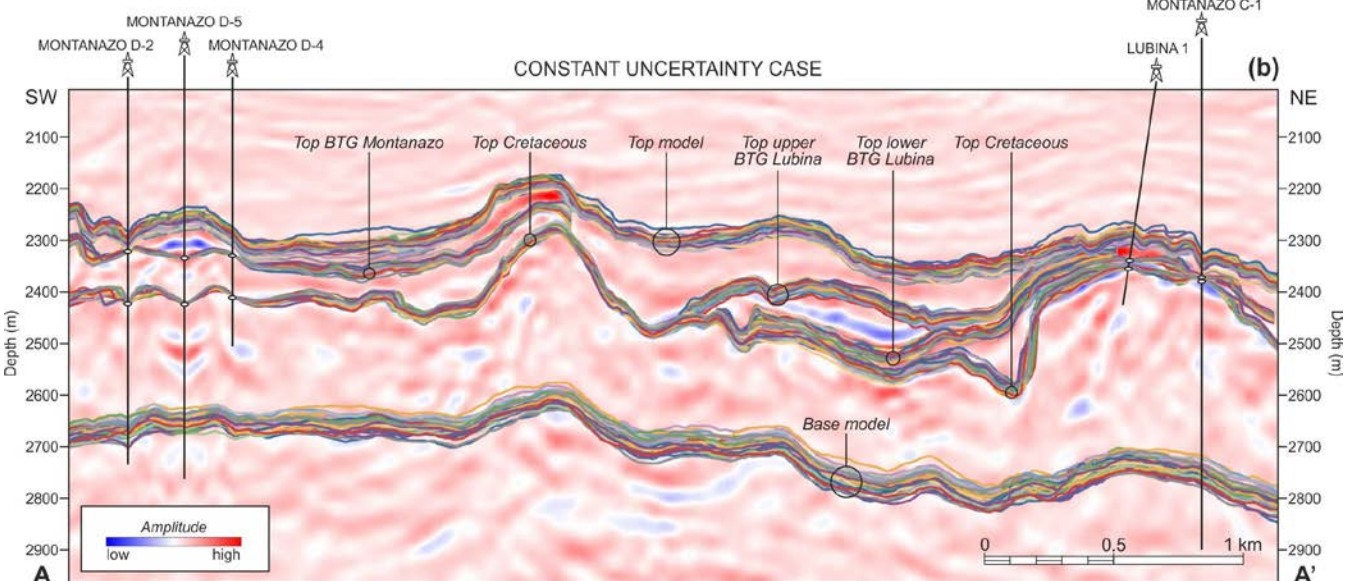

**Figure 16: Multiple realizations of the horizons modeled by capturing the uncertainty using the manual and the constant uncertainty cases. Only 50 of the 200 realizations are shown in each case. Note that the uncertainty is nonexistent in the well positions for those horizons with well picks and that the horizons showing the maximum spread are those without well pick data available. It can also be appreciated that the spread of uncertainty is visually higher in the constant than in the manual uncertainty cases, and that in both cases increases with the depth of the horizon.**