# Peer review of "Quantifying the impact of the structural uncertainty on the gross rock volume in the Lubina and Montanazo oil fields (Western Mediterranean)"

_Solid Earth, 2019_

## Referee Comment (RC1) · Vedad Hadziavdic (Referee) · 29 May 2019

GENERAL COMMENTS

Awareness about significance of structural uncertainty in the subsurface models and its effects on costly decisions has risen sharply in the oil industry over the last decade. Commercial tools to model and simulate structural uncertainty have become easily available, either as stand-alone applications or fully integrated in widely used reservoir modeling software packages. However, understanding how the uncertainties affect the

results, how to parametrize the models (determine priors, trends, variograms), or how to use results (which are probability distributions) remains a challenge. The bridge between statistics on one hand and geophysics and geology on the other is still not easy to cross for a typical geoscientist.

The paper touches upon some of these issues in an interesting case-study. It is nicely structured. It starts with a clear motivation and a nice overview over the most important elements needed to be considered. It continues with a description of the geological setting for the reservoir.

The dataset if described very shortly and lacks some important information, which should be included when assessing the uncertainties discussed in the paper.

The paper continues to discuss specifics of the uncertainty modelling for both horizons and the faults. Fault uncertainty modelling is not often included in the structural modelling, due to several reasons. The authors demonstrate the importance of fault uncertainty on the observed differences in GRV. In the interpretation of the results, the authors show how structural uncertainty affects GRV and distribution of oil in the reservoir. They use clearly the cross-sections of the reservoir, histograms and box-plots to describe the effects of structural uncertainty.

What lacks in the paper is a more detailed discussion about two important issues.

The first one is related to the specification of the interpretation uncertainty. What type of seismic survey is used? How is the frequency content related to the width of the envelopes used in manual uncertainty picking? What is the main source of uncertainty in picking top reservoir?

The second issue is related to what affects uncertainty in top structure. It is a combined effect of uncertainty in velocity model, uncertainty in well picks and seismic interpretation. For a reservoir at this depth (time), depending on the choice of velocity model, velocity uncertainty could have a much bigger impact on the structural uncertainty than

time interpretation. By including only one, and leaving all others out, what may happen is often called "ballooning effect". All the other uncertainties are squeezed and their effect pops out in an ambiguous way within the time interpretation uncertainty. This might lead to overestimating the effect of picking uncertainty on the top structure.

The paper is however interesting and shads light on an important topic which needs more investigation.

SPECEFIC COMMENTS

In Section 3.1. the authors introduce uncertainty envelopes around initial horizons. They base the uncertainty on variations in diffractions and amplitude of the data. Judging from the example and illustration in Figure 4., diffractions do not seem to play a role in the picking uncertainty. In general, it is difficult to see clear footprint of diffractions (smiles) in the seismic presented in the work. What seems to guide uncertainty picking in Figure 4. is frequency of the seismic event (Top lower BTG Lubina). The lower the frequency, the higher picking uncertainty is introduced.

It is not argued why this should be the case. In broadband surveys, which contain more low-frequency information, the seismic image displays wider reflection "bands". At the same time, increased low frequency content provides higher resolution in the seismic. This means that picking visually might seem more ambiguous. At the same time, the increased resolution will improve detection of elastic changes in the subsurface.

A counter example is a conventional survey with narrow frequency band. Such a survey will suffer from dominant side lobes in the wavelet. Visually, they seem to provide higher frequency content and narrow reflection events which are easy to pick. Typically, a geologist will prefer to pick on such seismic. However, this is an erroneous assumption, which will lead to overconfidence in picking based on events which do not represent real elastic changes in the subsurface.

More information about the seismic survey, tuning effects and frequency content should

be provided to be able to specify picking uncertainty in a more confident way. As it is demonstrated in the paper, these prior uncertainties are very important away from the wells.

In Section 3.3. the prior model is presented in detail. For the spatial continuity of the residuals, spherical variogram is chosen, which is reasonable choice for this kind of applications. However, the variogram range was set to 500m, based on the argument that it should not be more than half the reservoir width (Section 5.3). The expected lateral continuity of the geological layering is mentioned but not further discussed. The range of 500m is very short for most of the structural settings. The kriged depth surfaces will be uncorrelated for distances larger than 500m. However, already at much shorter distances (e.g. 250m), spherical variogram will only require weak correlations. If the prior uncertainties are large, and there are few data points in the data set, the simulated surfaces will exhibit large depth fluctuations over short distances. On one hand, it is geologically questionable. On the other, it will lead significant local changes in GRV which might not be realistic or informative.

This is to some degree illustrated in Figure 6 (b) where significant changes in depth over short distances can be observed in some areas, which apparently are not related to fault transitions.

As the uncertainty in velocity model could strongly dominate the total uncertainty in the structure, the model needs to be presented and some discussion needs to included. The uncertainty in depth can be roughly described as $d(Z)=d(V*t)=dV*t + V*dt$. Considering time of the event, average velocity and uncertainty in both variables, the effects of the two can be roughly compared.

Quite often, changing the time interpretation does not change the depth surface significantly, given a large uncertainty in the velocity.

---

## Referee Comment (RC2) · Alexander Schaaf (Referee) · 4 Jun 2019

General comments

The paper presents an interesting case study of the impact of structural uncertainties on structural geomodels. The authors compare two different approaches of uncertainty parametrization of structural geomodels from seismic data and their impact on gross rock volume estimations. The manuscript is well structured, and both title and abstract are adequate for the content.

[Figure]

The manuscript gives a good introduction to the uncertainties involved in seismic interpretation and subsequent structural geomodeling and clearly states the motivation for and relevance of their work in the context of economic geology.

The methodology section and clarity of the paper would be significantly improved if the authors would elaborate more on the details of how they parametrized the stochastic geomodel. It is not very clear what kind of statistical distributions were used for the Monte Carlo simulation, which is important to interpret the results. More detail on this can be found in the specific comments below.

The results are clearly presented, and the authors give a good visual overview of the effects of uncertainty on the structural geomodel and GRV estimates.

The discussion is overall good but could use some minor improvements outlined in the specific comments below. It could also use a more detailed integration of the results in context of related work. The figures presented are overall good, but the use of a perceptually uniform colormap is recommended (see technical comments below).

Overall, I believe this manuscript to be a good submission for the special issue on uncertainty in the geosciences. The scientific quality of the manuscript is good but should be further improved regarding the stochastic parametrization.

Specific comments

P6L25 – The sentence is not clear to me: what exactly is meant by geophysical data? This should be more specific for the reader.

P6L25-26 – The sentence is unclear. As I interpret it, the authors use the base case model and modify it using samples from perturbance distributions for faults and horizons to create new structural geomodels. I recommend specifying this.

P7L10-12 – What does "constant envelope" mean in context of the stochastic simulation? A Uniform distribution with the stated bounds? I recommend the authors expand this a bit to clearly state the stochastic parametrization of their uncertainty model.

P7L12-13 – How exactly was the envelope adjusted to the seismic quality and what exactly does seismic quality refer to in this context?

P8L10 – This sentence needs to be corrected. A stochastic simulation can only create equi-probable realizations if only Uniform distributions were used. The authors used Gaussian Random Fields and Gaussian distributions in their simulation (Fig. 7), and therefore the samples/realizations are not equi-probable. It is also unclear to me what "spans the uncertainty" means. A Monte Carlo simulation (or any stochastic simulation methods) will only ever reproduce the exact uncertainty in limit to infinite samples.

P8L24 – Why 200 realizations? Stochastic simulations need to balance computational cost with representative sampling, and the number of samples is critical for an accurate representation of the uncertainty within the results. I recommend the authors elaborate why they chose this number.

P11L7-9 – I recommend comparing Inner Quartile Ranges instead of minimum and maximum values to describe and compare the uncertainty. Minimum and maximum values are not necessarily representative.

P11L15 – Parametrization relies on assumptions, thus the word "accurate" can be misleading here. There is ample room for human bias and error in interpreting uncertainty from seismic data.

P11L25 – It is unclear to me what exactly is compared here. The mean GRV values of the different simulations?

P12L24 – Why are trend uncertainty values typically up to 10% of the depth of the horizon? Is this a rule of thumb, is there actual data on this? The source of this information should be clarified.

P12L13 – Why should the variogram range for the residual uncertainty "in general not be more than the half of the reservoir size"? It is unclear if this is a rule of thumb or based on actual studies. This should be clarified.

P13L3-5 – Manual interpretation of seismic data is prone to human error and bias. This should be discussed in this paragraph.

Technical corrections

Figure 13d – It's really hard to see any differences in this plot due to the small size and scale of the y-axis and use of arrow heads.

Figures 6, 8, 9, 12 – All figures make us of rainbow color schemes, which is perceptually not uniform. This makes it more difficult for readers to correctly perceive the underlying data. I highly recommend use of a perceptually uniform colormaps which are also more robust to color blindness.

---

## Author Comment (AC1) · 1 Jul 2019

Dear Vedad Hadziavdic,

We much appreciate your revision of our manuscript. Your constructive comments and suggestions have helped us to clarify some key points of the paper and to improve it. We have prepared a detailed response to your comments with the specification of all the changes made on the manuscript, which will be uploaded along with the revised version of the manuscript.

[Figure]

The dataset section has been completed by describing the seismic survey and the velocity model used. We have also completed the discussion section as suggested, by mainly including some discussion about seismic frequency content, the "ballooning effect" and the uncertainty in depth associated with the velocity model. Finally, we have also addressed the issue related to the spatial continuity of the residuals and the variogram range. We hope that the revisions made meet the important points raised.

Please also note the supplement to this comment:
https://www.solid-earth-discuss.net/se-2019-64/se-2019-64-AC1-supplement.pdf

[Figure]

**Supplement:**

**AUTHOR RESPONSE TO RC1 (Vedad Hadziavdic)**

We try to summarize below the referee main comments and the authors response, along with the actions and changes done to improve the previous manuscript. We have also clarified some points when needed.

**GENERAL COMMENTS:**

*"Awareness about significance of structural uncertainty in the subsurface models and its effects on costly decisions has risen sharply in the oil industry over the last decade. Commercial tools to model and simulate structural uncertainty have become easily available, either as stand-alone applications or fully integrated in widely used reservoir modeling software packages. However, understanding how the uncertainties affect the results, how to parametrize the models (determine priors, trends, variograms), or how to use results (which are probability distributions) remains a challenge. The bridge between statistics on one hand and geophysics and geology on the other is still not easy to cross for a typical geoscientist. The paper touches upon some of these issues in an interesting case-study. It is nicely structured. It starts with a clear motivation and a nice overview over the most important elements needed to be considered. It continues with a description of the geological setting for the reservoir.*

*The dataset if described very shortly and lacks some important information, which should be included when assessing the uncertainties discussed in the paper.*

*The paper continues to discuss specifics of the uncertainty modelling for both horizons and the faults. Fault uncertainty modelling is not often included in the structural modelling, due to several reasons. The authors demonstrate the importance of fault uncertainty on the observed differences in GRV. In the interpretation of the results, the authors show how structural uncertainty affects GRV and distribution of oil in the reservoir. They use clearly the cross-sections of the reservoir, histograms and box-plots to describe the effects of structural uncertainty.*

*What lacks in the paper is a more detailed discussion about two important issues.*

*The first one is related to the specification of the interpretation uncertainty. What type of seismic survey is used? How is the frequency content related to the width of the envelopes used in manual uncertainty picking? What is the main source of uncertainty in picking top reservoir?*

*The second issue is related to what affects uncertainty in top structure. It is a combined effect of uncertainty in velocity model, uncertainty in well picks and seismic interpretation. For a reservoir at this depth (time), depending on the choice of velocity model, velocity uncertainty could have a much bigger impact on the structural uncertainty than time interpretation. By including only one, and leaving all others out, what may happen is often called "ballooning effect". All the other uncertainties are squeezed and their effect pops out in an ambiguous way within the time interpretation uncertainty. This might lead to overestimating the effect of picking uncertainty on the top structure.*

*The paper is however interesting and shads light on an important topic which needs more investigation."*

**Authors comment:** We are very grateful for your general comments and suggestions. We agree with the reviewer that the dataset should be presented with more details and that the discussion should be completed with regards to the relationship between the uncertainty envelopes and the

type of seismic data used. It is also important to comment that the effect of the interpretation uncertainty in picking the horizons and faults could be overestimated if the others uncertainties are not included (i.e. velocity and well pick uncertainties).

Following these comments, we have tried to improve the manuscript by completing the description of the dataset and also the discussion section:

- o Regarding the dataset, the seismic data used in this work is a pre-stacked time migrated seismic cube. It was acquired and processed in 2006. A conventional acquisition system with a bandwidth between 8 – 80 Hz was used. The frequency content ranges from 20 to 40 Hz and the time window for the seismic picking is 20 ms. The vertical resolution of the seismic data is 45 m.
  The velocity model for the time-to-depth conversion was built using a 3D root-mean-square migration velocity field that was calibrated with velocities from wells (using a sonic log and a check-shot survey). Interval velocities ranged from 1528 m/s to 5028 m/s.

  This information has been included in the corresponding Dataset section of the revised version of the manuscript.

- o Regarding the two issues to be discussed, we have added a new discussion section ("5.3 Seismic survey and other sources of structural uncertainty"), where we have commented the relationship between the envelopes and the frequency content, and the importance of capturing the velocity uncertainty.

**SPECIFIC COMMENTS:**

**Specific comment #1:** *"In Section 3.1. the authors introduce uncertainty envelopes around initial horizons. They base the uncertainty on variations in diffractions and amplitude of the data. Judging from the example and illustration in Figure 4., diffractions do not seem to play a role in the picking uncertainty. In general, it is difficult to see clear footprint of diffractions (smiles) in the seismic presented in the work. What seems to guide uncertainty picking in Figure 4. is frequency of the seismic event (Top lower BTG Lubina). The lower the frequency, the higher picking uncertainty is introduced.*

*It is not argued why this should be the case. In broadband surveys, which contain more low-frequency information, the seismic image displays wider reflection "bands". At the same time, increased low frequency content provides higher resolution in the seismic. This means that picking visually might seem more ambiguous. At the same time, the increased resolution will improve detection of elastic changes in the subsurface.*

*A counter example is a conventional survey with narrow frequency band. Such a survey will suffer from dominant side lobes in the wavelet. Visually, they seem to provide higher frequency content and narrow reflection events which are easy to pick. Typically, a geologist will prefer to pick on such seismic. However, this is an erroneous assumption, which will lead to overconfidence in picking based on events which do not represent real elastic changes in the subsurface.*

*More information about the seismic survey, tuning effects and frequency content should be provided to be able to specify picking uncertainty in a more confident way. As it is demonstrated in the paper, these prior uncertainties are very important away from the wells."*

**Authors comments:** Thank you for your comment. We totally agree with you in that we have not commented properly how the manual uncertainty envelops have been defined and that

discussing the relationship between the frequency content and the uncertainty envelopes has not been sufficiently considered in the earlier version of the manuscript.

Different actions in the revised version of the manuscript have been done:

- o We have specified that the size of the envelopes in the manual uncertainty case vary according to frequency content events, lateral variations of amplitudes along reflectors, diffraction areas and how the reflectors terminate around fault surfaces when fault reflections are not present in the seismic image. This has been changed in the Abstract (P1L21-23); in section "3.1 Manual and constant uncertainty workflows" (P6L14-15 and P6L19-23); in section "3.2 Fault uncertainty modeling" (P7L20-23); in section "3.3 Horizon uncertainty modeling" (P8L7-8); and in section "6 Conclusions" (P16L1-2).
- o Some discussion has been added in new section "5.3 Seismic survey and other sources of structural uncertainty".

*Specific comment #2:* "*In Section 3.3. the prior model is presented in detail. For the spatial continuity of the residuals, spherical variogram is chosen, which is reasonable choice for this kind of applications. However, the variogram range was set to 500m, based on the argument that it should not be more than half the reservoir width (Section 5.3). The expected lateral continuity of the geological layering is mentioned but not further discussed. The range of 500m is very short for most of the structural settings. The kriged depth surfaces will be uncorrelated for distances larger than 500m. However, already at much shorter distances (e.g. 250m), spherical variogram will only require weak correlations. If the prior uncertainties are large, and there are few data points in the data set, the simulated surfaces will exhibit large depth fluctuations over short distances. On one hand, it is geologically questionable. On the other, it will lead significant local changes in GRV which might not be realistic or informative.*

*This is to some degree illustrated in Figure 6 (b) where significant changes in depth over short distances can be observed in some areas, which apparently are not related to fault transitions.*"

Authors comments: Thanks for this comment. We think that some clarification to this point is needed.

The variogram range does not control the correlation distance of a simulated horizon independently of other parameters like the trend uncertainty, but accounts for the correlation of small-scale variations in the depth of the horizon. These small-scale local variations are associated with differences between the position of the trend and the well data, and also with the frequency content, the amplitude of the reflectors and the presence of diffraction zones that make the picking of the horizons uncertain.

The residual is described by two components: vertically, by the standard deviation maps generally derived from the interpreted envelopes in the seismic interpretation, and laterally, by the variogram. Therefore, the variogram range only represents the horizontal continuity of the small-scale variations of the simulated horizons.

The trend controls the large-scale variations. The final position and geometry of a simulated horizon results from the sum of the trend and the residual. This is illustrated in fig. 6. In the example of this figure, the horizon represented in (c) is a simulated horizon corresponding to the sum of the trend shown in (a) and the residual shown in (b). As it can be noted, the depth and the general geometry of the simulated horizon is very similar to those of the trend, indicating that the trend uncertainty exerts an important control on the simulated horizons. In the example shown in fig. 6, the residual represent variations in the horizons depth of about 0.05%.

To clarify this in the text, we have replace the sentence "represent the expected lateral continuity of the geological layering" in line 31 of page 12 in the original manuscript by "accounts for the correlation of small-scale variations in the depths of the horizons" in the revised version of the manuscript.

*Specific comment #3:* *"As the uncertainty in velocity model could strongly dominate the total uncertainty in the structure, the model needs to be presented and some discussion needs to included. The uncertainty in depth can be roughly described as d(Z)=d(V\*t)=dV\*t + V\*dt. Considering time of the event, average velocity and uncertainty in both variables, the effects of the two can be roughly compared.*

*Quite often, changing the time interpretation does not change the depth surface significantly, given a large uncertainty in the velocity.*

Authors comments: As commented in the manuscript, the objective of this paper is to capture the uncertainty in the seismic interpretation, although we agree with the referee that other uncertainties like the uncertainty in the velocity model or in the well data are also important structural uncertainties that deserve much attention. Following your recommendation, we have presented the velocity model in section "1.2 Dataset" (as commented in the Authors response to the General comments). We have also completed the discussion section by including some estimations of the velocity uncertainty, comparing them with the uncertainty in picking the horizons and by commenting the importance of investigating the uncertainty in the velocity model in future works. This part of the discussion has been linked to that of the "ballooning effect" (see new discussion section "5.3 Seismic survey and other sources of structural uncertainty").

---

## Author Comment (AC2) · 1 Jul 2019

Dear Alexander Schaaf,

We much appreciate your revision of our manuscript. We have considered your constructive comments and suggestions to try to clarify and improve the manuscript, especially in relation with the parametrization of the stochastic model. We have addressed all the specific comments in the detailed response that will be uploaded, where we also specify the changes made in the manuscript. We hope that the revision

made gives enough detailed to the methodology for the uncertainty modeling, and addresses the important points raised.

Please also note the supplement to this comment:
https://www.solid-earth-discuss.net/se-2019-64/se-2019-64-AC2-supplement.pdf

[Figure]

**Supplement:**

**AUTHOR RESPONSE TO RC2 (Alexander Schaaf)**

We summarize below the referee comments and the authors response, along with the actions and changes done to improve the manuscript.

**GENERAL COMMENTS:**

*"The paper presents an interesting case study of the impact of structural uncertainties on structural geomodels. The authors compare two different approaches of uncertainty parametrization of structural geomodels from seismic data and their impact on gross rock volume estimations. The manuscript is well structured, and both title and abstract are adequate for the content.*

*The manuscript gives a good introduction to the uncertainties involved in seismic interpretation and subsequent structural geomodeling and clearly states the motivation for and relevance of their work in the context of economic geology.*

*The methodology section and clarity of the paper would be significantly improved if the authors would elaborate more on the details of how they parametrized the stochastic geomodel. It is not very clear what kind of statistical distributions were used for the Monte Carlo simulation, which is important to interpret the results. More detail on this can be found in the specific comments below.*

*The results are clearly presented, and the authors give a good visual overview of the effects of uncertainty on the structural geomodel and GRV estimates.*

*The discussion is overall good but could use some minor improvements outlined in the specific comments below. It could also use a more detailed integration of the results in context of related work. The figures presented are overall good, but the use of a perceptually uniform colormap is recommended (see technical comments below).*

*Overall, I believe this manuscript to be a good submission for the special issue on uncertainty in the geosciences. The scientific quality of the manuscript is good but should be further improved regarding the stochastic parametrization."*

**Authors comment:** Thank you very much for your comments and suggestions, especially for those regarding the parametrization of the stochastic uncertainty modeling. We have considered your recommendations to try to improve the clarity and the quality of the manuscript.

In the list below, we specify the actions done after each of your comments addressed.

**SPECIFIC COMMENTS:**

*P6L25* – The sentence is not clear to me: what exactly is meant by geophysical data? This should be more specific for the reader.

**Authors comment:** We agree with the referee and we have changed the text by replacing "geophysical data" by "picking the seismic events (faults and the horizons)". We have combined this sentence with the following sentence (see the P6L25-26 Referee comment just below) to improve in clarity.

*P6L25-26* – The sentence is unclear. As I interpret it, the authors use the base case model and modify it using samples from perturbance distributions for faults and horizons to create new structural geomodels. I recommend specifying this.

Authors comment: Following the recommendation of the referee, we have clarified the original sentence as follows (also including the sentence of the comment above):

"The base case model (in time or in depth domain, depending on the modeling workflow) is a reference to subsequently build new structural models using the sampled perturbance distribution for the faults and horizons. Perturbances record the structural uncertainty in picking the seismic events (faults and the horizons)." (P6L28-31).

*P7L10-12* – What does "constant envelope" mean in context of the stochastic simulation? A Uniform distribution with the stated bounds? I recommend the authors expand this a bit to clearly state the stochastic parametrization of their uncertainty model.

Authors comment: Constant envelope means that the dimension of the uncertainty zone around each fault or horizon interpretation is set as a constant distance to that interpretation (i.e. constant distance above and below the horizons, and along both sides of the faults), independently of the fact that the uncertainty area in picking the surface varies across the interpreted trace.

We have clarified this in section "3.1 Manual and constant uncertainty workflows" (P6L19-23).

*P7L12-13* – How exactly was the envelope adjusted to the seismic quality and what exactly does seismic quality refer to in this context?

Authors comment: The envelope in the manual case was adjusted according to the extent of the uncertainty area (i.e. zone of ambiguity in picking a surface) along the interpreted surface. This has been clarified as follows (P7L20-23):

"In the interpreted case, this envelope was adjusted according to how the uncertainty zone (i.e. zone of ambiguity when picking the fault) varies along each interpreted fault trace. In the study dataset, faults do not produce seismic reflections nor diffractions and their interpretation was mainly based on the distance of the reflector terminations (Fig. 4)."

Additionally, we have modified the text in section "3.1 Manual and constant uncertainty workflows" (P6L14-15) concerning the ambiguity zones in picking horizons and faults. These zones represent the uncertainty in the interpretation of a surface related to low frequency content events, lateral variations of amplitudes along reflectors, diffraction areas or the lack of seismic reflections of faults.

*P8L10* – "This sentence needs to be corrected. A stochastic simulation can only create equi-probable realizations if only Uniform distributions were used. The authors used Gaussian Random Fields and Gaussian distributions in their simulation (Fig. 7), and therefore the samples/realizations are not equi-probable. It is also unclear to me what "spans the uncertainty" means. A Monte Carlo simulation (or any stochastic simulation methods) will only ever reproduce the exact uncertainty in limit to infinite samples.

**Authors comment:** Following the comment of the referee, we have modified the sentence in the revised version of the manuscript as follows (P8L20):

"The stochastic simulation produces multiple realizations of the horizons, which represent the input uncertainty."

*P8L24* – Why 200 realizations? Stochastic simulations need to balance computational cost with representative sampling, and the number of samples is critical for an accurate representation of the uncertainty within the results. I recommend the authors elaborate why they chose this number.

**Authors comment:** We have completed the text in new version of the manuscript explaining why 200 realizations were obtained (P9L1-5): "With this number of realizations, the fluctuations in the predicted GRV stabilizes resulting in the output statistics being enough representative of the uncertainty captured."

*P11L7-9* – I recommend comparing Inner Quartile Ranges instead of minimum and maximum values to describe and compare the uncertainty. Minimum and maximum values are not necessarily representative.

**Authors comment:** Following this comment and according to the statistical data and figures in the paper, we present the ranges corresponding to the difference between the P10 and P90 (instead of between minimum and maximum values) in the revised version of the manuscript. Changes in the text are in section "4.2 GRV in the prediction models and in the simulated realizations" (P10L26-29); and in section "5.1 Manually interpreted versus constant uncertainty cases" (P11L19-20). Additionally, we also comment the percentage differences of P10 and P90 realizations with respect to the base case model instead of maximum and minimum values (see changes in the "Abstract" section in the revised manuscript (P1L31).

*P11L15* – Parametrization relies on assumptions, thus the word "accurate" can be misleading here. There is ample room for human bias and error in interpreting uncertainty from seismic data.

**Authors comment:** We agree with the Referee comment. We have deleted this term from the corresponding sentence in the new version of the manuscript (P11L28-29).

*P11L25* – It is unclear to me what exactly is compared here. The mean GRV values of the different simulations?

**Authors comment:** We compared the difference in percentage between the maximum GRV and the base case, and the minimum GRV and the base case for the three OWC, and separately in the manual and the constant cases. According to the Referee comment of P11L7-9, we use the P10 and P90 values instead of maximum and minimum values in the revised version of the manuscript. Thus, the corresponding percentages presented in the reviewed manuscript have changed (see P1L31 in the "Abstract" section; and P12L5-7 in "5.2 Impact of the structural uncertainty in a small and fractured reservoir").

*P12L24* – Why are trend uncertainty values typically up to 10% of the depth of the horizon? Is this a rule of thumb, is there actual data on this? The source of this information should be clarified.

**Authors comment:** This percentage was provided by Emerson Automation Solutions, but there is no specific reference to document this. A personal communication reference has been added to the revised manuscript.

*P12L13* – Why should the variogram range for the residual uncertainty "in general not be more than the half of the reservoir size"? It is unclear if this is a rule of thumb or based on actual studies. This should be clarified.

**Authors comment:** This is the same case than the comment above. A personal communication reference was used in the revised manuscript.

*P13L3-5* – Manual interpretation of seismic data is prone to human error and bias. This should be discussed in this paragraph.

**Authors comment:** We have added a comment on human error in interpretation in the corresponding paragraph (P14L33-P15L1).

*TECHNICAL CORRECTIONS:*

*Figure 13d* – It's really hard to see any differences in this plot due to the small size and scale of the y-axis and use of arrow heads.

**Authors comment:** Following the comment of the referee, we have modified fig. 13d, and accordingly, also figs. 14d and 15d. We have duplicated the vertical scale to see more easily differences in these plots. The vertical exaggeration (2×) has also been indicated in the corresponding figure captions.

*Figures 6, 8, 9, 12* – All figures make us of rainbow color schemes, which is perceptually not uniform. This makes it more difficult for readers to correctly perceive the underlying data. I highly recommend use of a perceptually uniform colormaps which are also more robust to color blindness.

**Authors comment:** We thank the suggestion of the referee and agree than in some cases using perceptually unifom colormaps is an appropriate choice. However, we have selected the same color scale for some of the maps and images in the commented figures in order to make possible a direct visual comparison of the results.

In the case of fig. 6, the residual map is shown with a specific color table, different from the one for the trend and the simulated horizon, which both show the same range of values.

In fig. 8, we selected different color tables for the manual and for the constant uncertainty cases. Amongst the maps of each case, we use the same color table in order to highlight that for those horizons without well picks available (e.g. Top model, Top upper BTG Lubina, Base model) higher prediction errors are recorded than for those horizons with well picks. This has been also commented in the new version of the manuscript (P11L32-33 in section "5.1 Manually interpreted versus constant uncertainty cases" and in the caption of figure 8). Additionally, using the same color table allows comparing the degree of lateral variation in the prediction error in each horizon and for each modeling case.

In Fig. 9a and b, the color table is showing the depths of all the horizons, so there is no possibility of using perceptually uniform colormaps. In Fig. 12, the color scale is showing the GRV in each

modeled cell above the oil-water contact. In this image, we consider that comparing the variation of the oil zone extent amongst the different scenarios shown is the most relevant objective, and that using the same color tables helps in direct comparison of results.

---

## Author Comment (AC3) · 1 Jul 2019

Please, Find in this folder the revised version of the manuscript.

Please also note the supplement to this comment:
https://www.solid-earth-discuss.net/se-2019-64/se-2019-64-AC3-supplement.zip

---

## Author Response (AR2)

**AUTHORS RESPONSE TO THE TOPICAL EDITOR**

We have revised the manuscript considering all your comments. Below we specify the changes made. We also include the marked-up manuscript version showing the changes made after the list of comments and changes.

*Comment #1:* *Please provide a full definition for GRV in the abstract.*

**Changes made:** We have replaced "GRV" by "gross rock volume" in the abstract. Then, in section "1 Introduction", we have defined the term as follows:

Page 4, lines 9 – 10: "… gross rock volume (GRV), i.e. volume of the reservoir rock above the oil-water contact".

*Comment #2:* *Pg 6 line 19 Can you be really clear how you made the manual uncertainty envelopes, was this a visual inspection or did you use the seismic data in anyway?*

**Changes made:** We have changed the sentence to clarify how we defined the manual uncertainty envelopes as follows:

Page 6, lines 19 – 23: "In the manual case, the dimensions of the uncertainty envelopes around each interpreted surface are based on a visual inspection of variations in the frequency content and in the amplitude of the seismic event, of the distance of reflector terminations around faults and of diffractions, if present."

*Comment #3:* *Pg 9 line 4 delete 'enough'*

**Changes made:** We have deleted 'enough'.

*Comment #4:* *Pg 13 line 1 'through' perhaps should be trough, but this is not clear. Please check this and through to line 5 for clarity.*

**Changes made:** We have replaced 'through' by 'trough' (page 12, line 23) and 'pick' by 'peak' (page 12, line 24).

*Comment #5:* *Pg 14 lines 1-9 Please check this new section on depth conversion for clarity/English. You show that the depth conversion is critical and has a bigger impact on the uncertainty than the interpretation uncertainty modelling undertaken. This leads to the question of why you did not include this in the first place. I suggest you deal with this by describing how you would include it in a future improved workflow.*

**Changes made:** We have changed the final part of Section "5.3 Seismic survey and other sources of structural uncertainty" to improve clarity. Additionally we have also added a brief description of the workflow that could be developed to combine all sources of the structural uncertainty in the same workflow (page 13, lines 11 – 33, and page 14, lines 1 – 12).

**Other changes made:**

o   The reference Schaaf and Bond (in this volume) has been updated with the corresponding full reference (page 15, line 3, and page 20, lines 2 – 3).
o   The format of some references in the References section has been revised.

[revised manuscript text omitted]